# Disorders of Sex Development: Classification, Review, and Impact on Fertility

**DOI:** 10.3390/jcm9113555

**Published:** 2020-11-04

**Authors:** Pedro Acién, Maribel Acién

**Affiliations:** 1Department of Gynecology, Miguel Hernández University, San Juan Campus, 03550 San Juan, Alicante, Spain; macien@umh.es; 2Obstetrics and Gynecology, San Juan University Hospital, San Juan Campus, 03550 San Juan, Alicante, Spain

**Keywords:** sex development, anomalies, classification, gonadal dysgenesis, testicular dysgenesis, pseudohermaphroditism, ovotesticular disorders, congenital adrenal hyperplasia, adrenogenital syndrome, fertility

## Abstract

In this review, the elements included in both sex determination and sex differentiation are briefly analyzed, exposing the pathophysiological and clinical classification of disorders or anomalies of sex development. Anomalies in sex determination without sex ambiguity include gonadal dysgenesis, polysomies, male XX, and Klinefelter syndrome (dysgenesis and polysomies with a female phenotype; and sex reversal and Klinefelter with a male phenotype). Other infertility situations could also be included here as minor degrees of dysgenesis. Anomalies in sex determination with sex ambiguity should (usually) include testicular dysgenesis and ovotesticular disorders. Among the anomalies in sex differentiation, we include: (1) males with androgen deficiency (MAD) that correspond to those individuals whose karyotype and gonads are male (XY and testes), but the phenotype can be female due to different hormonal abnormalities. (2) females with androgen excess (FAE); these patients have ovaries and a 46,XX karyotype, but present varying degrees of external genital virilization as a result of an enzyme abnormality that affects adrenal steroid biosynthesis and leads to congenital adrenal hyperplasia; less frequently, this can be caused by iatrogenia or tumors. (3) Kallman syndrome. All of these anomalies are reviewed and analyzed herein, as well as related fertility problems.

## 1. Introduction

Disorders of sex development (DSDs) were defined as “congenital conditions within which the development of chromosomal, gonadal, and anatomic sex is atypical” at the Chicago Consensus Meeting in 2005, which was then published as a Consensus Statement in 2006 [1,2,3]. However, there have been and are many controversies around DSDs, both because of the negative connotations perceived by organizations and professionals when using terms such as “disorders,” and because some health professionals consider as inaccurate, non-descript, and confusing certain terms such as “intersex,” “pseudohermaphroditism,” “hermaphroditism,” and “sex reversal” [3,4,5]. Other authors [6,7] prefer the term “differences” and use the same acronym—DSD— to refer to differences of sex development; meanwhile, others, as well as ourselves, do not consider the nomenclature and classification proposed in the Chicago consensus as the most appropriate. We believe that it is preferable to analyze normal or physiological sex determination and subsequent sex differentiation to expose any anomalies, as they would be included in a pathophysiological and clinical classification of anomalies in sex development (ASDs) [8,9], and then to continue with their orderly exposure and with the related fertility problems.

## 2. Determination and Differentiation of the Human Sex

The phenotypic sex of a person depends on the type of gonad that develops in the embryo, a process that in itself is determined by the constitution or genetic inheritance of the individual, although the development of the gonads is different from that of any other organ, since they have the potential to differentiate into two functionally distinct organs, i.e., testes or ovaries [10,11]. However, there are many more elements that determine the sex of an individual; additionally, a person’s sexual identity includes any behaviors with sexual overtones, such as characteristic gestures and habits, ways of speaking, preferences about leisure, and the content of dreams. Indeed, hormonal influences not only affect internal and external genital development and differentiation; the embryo’s brain has also been shown to differentiate sexually [12,13,14], perhaps through control mechanisms similar to those developed by the external genitalia [15]. On the other hand, it is also possible that the induction of the central nervous system (CNS) by hormones affects the patterns of hormonal secretion and, consequently, sexual behavior in adults [16,17,18,19,20]. Thus, the sex of an individual and their sexual expression, i.e., homo- or heterosexual, must be considered the result of all influences that the individual receives, both before and after birth [8].

Sex development can be divided into two different processes: (1) the determination of sex, which is the destination of the undifferentiated gonad to form the testis or ovary, and which, as has been said, is a process that is genetically programmed in a critical way; (2) sex differentiation, which takes place through the hormones produced by the gonads both during their formation and once they have been formed. Both processes have been extensively analyzed in another publication [9]; thus, here, we only mention the constituent elements of both processes involved in human sex development, as shown in Figure 1.

Briefly, sex determination depends on the chromosomes and genes that an individual carries, and on the type of gonad that differentiates (gonadal differentiation), therefore including the chromosomal sex, the genetic sex, and the gonadal sex. The second process, known as sex differentiation, takes place once a decision regarding sex determination has been made, based on factors produced by the gonads that determine the development of the phenotypic sex. Therefore, sex differentiation includes the hormonal sex, the gonophoric sex (i.e., internal genitalia), the phenotypic sex (i.e., external genitalia and general phenotype), and the legal, educational, psychological, and social sex. Generally speaking, the factors influencing sex determination are transcriptional regulators, whereas the factors important for sex differentiation are secreted hormones and their receptors [10,11,21], although many of the genes responsible for sex determination are also responsible for gonadal steroidogenesis and, therefore, for subsequent sex differentiation. Disruption of any of the genes involved in these processes during the course of testicular or ovarian development, or in the subsequent sex differentiation, could lead to ASDs (or DSDs) [10,11].

## 3. Disorders of Sex Development: Classification

The most important milestones in the knowledge of DSDs or ASDs (previously called intersex states) could be the following:1876: Klebs made the first attempt at a classification, noting (1) true sexual ambiguity or true hermaphroditism (TH) and (2) pseudohermaphroditism (Ps): (a) male, if there were testicles, and (b) female, if there were ovaries [22].1931: Goldschmidt introduced the term intersex states and dropped homosexuality from his theory of intersexuality.1949: Baar discovered sex chromatin [23].1956: The karyotype was discovered and the chromosomes were typed.1960: Foss [24] published an article in Brit Med J on intersex states and made a controversial classification of them.1973–1975: The Y-linked histocompatibility (H-Y) antigen [25] was described and, later, the testis-determining factor (TDF) gene and the sex-determining region Y (SRY).2005–2006: The Chicago Consensus Meeting took place, in which the terms used on the subject were discussed, delegates considering them confusing or pejorative, and it was recommended to use the term “disorders of sex development” and a new classification proposal [3]. Dreger et al. [4] had already analyzed changing the nomenclature/taxonomy for intersex, calling for the abandonment of all terms based on the root “hermaphrodite.” Another expert consensus document was proposed in 2018, recommending the term “differences of sex development” and following a classification similar to that of Chicago, also based on the karyotype [6].

Recalling the points previously mentioned on the determination and differentiation of normal sex, we believe that the origin of abnormal sex development situations or ASDs must be: (1) in the chromosomes and SRY gene (and other sex-determining (SD) genes located in the autosomes) that may therefore be carried into the formation of the gonad; (2) in the hormone-producing glands involved in sex development (i.e., gonads, adrenals, and pituitary), as well as in the metabolization of these hormones, in their action on the target organs or in the content of receptors. Therefore, the classification that we must make regarding ASDs should be:Due to anomalies in the determination of the chromosomes and/or formation of the gonad; and,Due to anomalies in the differentiation of the genitalia and phenotype, which will be due to anomalies in hormonal secretions, or their action on the target organs, without anomalies in the chromosomes or in the gonads (at least from the point of view of the histological architecture) [26].

We, therefore, have proposed a new nomenclature [8,9] and suggest following a pathophysiological and clinical classification of ASDs, as shown in Table 1, based on the previously analyzed embryological and pathophysiological considerations [9(submitted)]. The corresponding nomenclature in the Chicago classification is also mentioned. Genitourinary malformations should not be included among the ASDs, since these are the consequence of abnormalities in the structural anatomical formation, during embryonic development, of the mesonephric (Wolffian ducts) and paramesonephric (Müllerian ducts) genital ducts in both sexes, including that of the urinary system [22,27,28,29,30,31,32].

Herein, we review and analyze each of the anomalies mentioned in the table and illustrate their impact on fertility.

## 4. DSDs due to Anomalies in Sex Determination (Chromosomes and/or Gonads)

### 4.1. Anomalies in Sex Determination without Sex Ambiguity (usually)

They include gonadal dysgenesis (i.e., Turner syndrome, mosaic variants, structural abnormalities of the second sex chromosome, and pure gonadal dysgenesis), polysomies, reversal sex, and Klinefelter syndrome. Both dysgenesis and polysomies with a female phenotype and the reversal sex and Klinefelter with a male phenotype. However, other situations of infertility related to oligo-azoospermia in males or primary ovarian insufficiency in females should also be included here as minor degrees of dysgenesis.

#### 4.1.1. Gonadal Dysgenesis

(1) True ovarian agenesis (gonadal agenesis) was described in 1944 by Wilkins and Fleischmann [33], and later, Linden and Overzier [34] evidenced its reality. There is no gonadal outline; the differentiation is female, the habit is childish, and they present primary amenorrhea with genital hypoplasia. There are no secondary sexual characteristics; the feeling is female, and the karyotype is 46,XX or 46,XY.

(2) However, gonadal dysgenesis (dysplasia, streak gonad [26]) is the arrest of gonadal development before differentiation into the testis or ovary, and the following must be distinguished: (1) gonadal dysgenesis with chromosomal alteration and in the body phenotype (i.e., Turner syndrome), and (2) pure gonadal dysgenesis, without chromosomal or phenotypic abnormality (i.e., Swyer syndrome).

Turner syndrome (TS) is the most common gonadal dysgenesis, affecting 1/2000 girls, the majority of which (> 95%) are infertile [35]. TS is characterized by the association of body phenotypic abnormalities (winged neck, cleft palate, short stature, sexual infantilism, shield chest, etc.) and by gonadal dysgenesis. The karyotype is 45,X0, that is, the second sex chromosome is missing, and the individual develops as a woman. However, among the autosomal genes of that second absent chromosome are the determinants of height and other organic functions (in the short arm X), in addition to the gonadal determining genes (in the long arm and proximal part of the short arm of X), since gonadal dysgenesis and phenotypic abnormalities are observed. Most die in utero (abortions with an X0 karyotype); only 3% survive, and of these, 90% present ovarian failure and primary amenorrhea. However, 10% of patients may have menstruations and even pregnancy, despite the X0 karyotype. Certainly, many of these cases will be mosaics, but there are numerous reports of TS with children and there are also well-documented risks for women with TS during pregnancy, including specific risks of aortic pathology, hepatic disease, thyroid disease, type 2 diabetes, and Cesarean section delivery [36]. Bernard et al. [37], when studying a large cohort of patients with TS (*n* = 480), observed that in most cases, primary ovarian failure occurred in prepuberty and spontaneous menarche was only reported in 20% of patients. Lymphedema on the back of the foot and hand is also characteristic and should alert to a diagnosis in newborns. However, it must be taken into account that not all symptoms are always present and that not all 45,X0 karyotypes, nor all of the phenotypic abnormalities mentioned, are exclusive to TS. That is, not all 45,X0 women have phenotypic abnormalities, nor are the phenotypic abnormalities mentioned unique to Turner; for example, they can also be seen in Noonan syndrome [38].

As mentioned, the majority of patients with TS are infertile. Spontaneous pregnancy is rare, occurring in only 2–5% of individuals, with an increased rate of fetal abnormalities and reproductive loss [39]. Infertility in these women is the result of an accelerated loss of oocytes that begins in fetal life and progresses during postnatal life [35], with the unique 45,X0 karyotype being predictive of gonads without the presence of follicles, while a mosaic karyotype and spontaneous menarche increase the likelihood of follicles [40]. However, despite the presence of follicles in these patients, their morphology is frequently abnormal, suggesting that the functional capacity of the oocytes may be affected [40]. In most people with TS, it is not possible to preserve fertility, and in these cases, fertility counseling should be offered and egg/embryo donation or adoption should be considered when appropriate [41]. Furthermore, given the frequent abnormal morphology of the follicles, the restoration of fertility by re-transplantation of cryopreserved ovarian tissue is unlikely to be successful, and, therefore, the subsequent use of cryopreserved ovarian tissue will likely require ex vivo oocyte maturation and artificial reproductive technologies. To date, there have been no reports of the restoration of fertility using cryopreserved ovarian tissues from individuals with TS [41].

(3) Ovarian dysgenesis or ovarian hypoplasia [22,42] is the most common form of hypoestronism or primitive ovarian failure. Here, the XX genetic sex coincides with the rudimentary ovary, but it is due to a gonadal disorder that was thought not to be dysgenetic but probably related to other SD genes (mutations involving the SRY-related gene SOX8) [43]. The following are distinguished: (1) afollicular ovarian dysgenesis that contains all ovarian structures but without follicles; (2) follicular ovarian dysgenesis, where there are primordial follicles but a lack of maturation; (3) ovarian hypoplasia. The symptoms are primary amenorrhea, infantile genitalia, moderately developed secondary sexual characteristics, habit, and female psychism. Hormone replacement therapy is recommended and the diagnosis is made by ovarian biopsy. The constitution XXX (47XXX) also presents as dysgenesis or ovarian hypoplasia to varying degrees, which sometimes results in fertility for women. The symptoms are, in general, primary amenorrhea or late menarche, infertility, normal phenotype, and mental retardation. However, polysomies or aneuploidies will be discussed in more detail later.

#### 4.1.2. Variants of Gonadal Dysgenesis

(1) Mosaic variants. Genetic mosaicism is responsible for most cases of gonadal dysgenesis variants. Mosaicism is the result of the development of two or more cell lines in an individual, and is usually the result of a mitotic error. The most common form of mosaicism with this syndrome, and which results in a positive chromatin sexual gonadal dysgenesis diagnosis, is the X0/XX mosaic. However, Lim et al. [44] described the incidence rate, puberty, and fertility in 45,X0/47,XXX mosaicism. They diagnosed a 10-year-old girl with TS with 45,X0/47,XXX mosaicism who presented with short stature but had spontaneous puberty, and pointed out that previous reports had demonstrated that TS with 45,X0/47,XXX is less severe than common TS due to higher occurrence of puberty (83%), menarche (57–67%), and fertility (14%) and a lower occurrence of congenital anomalies (< 5%). However, TS mosaicism may not reduce the frequency of short stature. In a letter to the editor on the case of Lim et al., Martin et al. [45] also reported another case with a predominance of line 47,XXX that did not have the clinical characteristics of Turner syndrome, instead having normal height and menarche at 14, but they did report the risk of premature ovarian failure.

(2) Structural abnormalities of the second chromosome. Remembering the location of the SD genes on the X or Y chromosomes, the consequences of these structural abnormalities can be well understood: gonadal dysgenesis and/or Turner-like abnormalities, even though the patient has all 46 chromosomes. A published case of Turner 46XXIq also had total gonadal and Müllerian agenesis [46].

Structural abnormalities of the Y chromosome with clinical impact are much less common than those of the X chromosome. Structural abnormalities include various types of translocations, inversions, deletions, duplications, and isochromosomes, and are estimated to affect less than 1% of the newborn male population, being particularly hazardous for male reproductive function [47]. Several patients with isochromosomes of the long arm of the Y chromosome (XYIq), isodicentric Y, and mosaics have been reported in boys with other congenital anomalies and girls with virilized external genitalia [48,49,50]. Such individuals are often phenotypic women with primary amenorrhea, sexual infantilism, and bilateral streak gonads. They have normal height and few of the somatic abnormalities associated with TS, but they do have varying degrees of lymphedema. Similarly, several patients have also been described with a dicentric chromosome for the long arm of the Y chromosome (XYdic) [51], and this is usually associated with mosaicism and with an X0 cell line. Thus, in this group of disorders, various phenotypes can be seen, including normal males, individuals with ambiguous genitalia, male Ps [52], and patients with many of the stigmata of gonadal dysgenesis.

(3) Pure gonadal dysgenesis. Pure gonadal dysgenesis means that there are no chromosomal or phenotypic abnormalities; only gonadal dysgenesis or “dysplasia” (dysplastic testis and streak gonad, as proposed by Lepais et al. [26]) is observed. The phenotype is normal female, being generally tall women. However, since the gonads are bands (streaks), they present primary amenorrhea and a certain sexual infantilism; the external genitalia are normal, as are the Müllerian ducts (i.e., uterus and tubes), although infantile. Naturally, they have elevated gonadotropins from gonadal failure. There are two main types of pure gonadal dysgenesis: One with a 46,XX karyotype [53], and another with a 46,XY karyotype, although there could be mosaics (which should not be included). Furthermore, the distinction is very important, because in XY patients with high frequency, they develop tumors, i.e., gonadoblastoma and sometimes dysgerminoma, which can even virilize [54]. Although there are also sometimes dysgerminomas in XX dysgenesis [55], this would depend on the dysplasia as reported by Lepais et al. [26]. Additionally, although some authors call all pure gonadal dysgenesis Swyer syndrome, really the term Swyer syndrome should be reserved for the XY gonadal dysgenesis described by Swyer in 1955: “Male pseudohermaphroditism, a hitherto undescribed form” [56]. Likewise, some authors have included cases with some sex ambiguity due to dysgenetic testes in Swyer syndrome, but these cases should be considered separately. Complete gonadal dysgenesis in 46,XY individuals results from mutations in the key testis-determining genes in a phenotypic female with internal Müllerian structures and bilateral streak gonads.

In summary, there is a group of patients who have streak gonads and persistent sexual infantilism, but who are of normal stature and in whom the other physical stigmata of TS are missing. Therefore, this disorder is referred to as pure gonadal dysgenesis. In such individuals, the eunuchoid habit is frequent and primary amenorrhea is the norm, although it is possible that they are taking the pill and have apparently normal cyclical menstruations and attend a consultationfor infertility. Gonadotropin concentrations increase at puberty as in prepubertal castrated individuals, and the patients can develop gonadal malignancies, such as dysgerminomas or gonadoblastomas, which should be taken into account in their conduction, as in all patients with stigma of gonadal dysgenesis in which the Y chromosome is present. From the point of view of fertility, the lack of either testicular or ovarian tissue means that it is not possible to obtain spermatozoa or oocytes from these individuals, and therefore, infertility is universal. However, the presence of Müllerian structures means that pregnancy may be possible using donor eggs or embryos and in vitro fertilization (IVF), as illustrated by a recent report of successful pregnancies in two sisters, with a healthy live birth in one and an ongoing pregnancy in the other [57].

Among our patients, there have been several cases of pure gonadal dysgenesis: (1) some were 46,XX, and in some cases, they presented with amenorrheic bumps and early ovarian failure (therefore indicating ovarian dysgenesis, leading early menopause) and the gonadal biopsy showed ovarian stroma without follicles. Other patients showed primary amenorrhea, gonadal dysgenesis, and, in one of them, vertebral malformations. (2) Another two cases were 46,XY (Swyer’s syndrome): the first patient wasmarried, with infertility and with streak gonads; the second single, and presenting bilateral microscopic gonadoblastoma. In any case, the gonadal bands (streaks or ‘cintillas’) should be removed and hormone replacement therapy given.

#### 4.1.3. Polysomies

Polysomies are normally associated with mosaic, but in cases with triplet X, they may not present abnormal sex development. XXY, XYY, or other polysomies with Y usually correspond to Klinefelter syndrome, as we will see later. However, Trisomy X is the most common female chromosomal abnormality, occurring in approximately 1 in every 1,000 female births. As some people are mildly affected or are asymptomatic, it is estimated that only 10% of people with trisomy X are diagnosed. As already mentioned, the most frequently reported condition in women with XXX aneuploidy is premature ovarian failure (POF), leading to early menopause, followed by habitual abortion. Tartaglia et al. [58] comprehensively reviewed Trisomy X and found that the most common physical features include tall stature, epicantal folds, hypotonia, and clinodactyly. Seizures, renal and genitourinary abnormalities, and POF may also be associated findings.

#### 4.1.4. Sex Reversal or Male XX

Sex reversal is an uncommon clinical syndrome described by De la Chapelle et al. in 1964 [59], and it occurs in 1/20,000 newborn males [60]. These individuals have a male sexual phenotype, with normal height and perhaps a certain degree of breast development (gynecomastia). The testicles are housed in the scrotal bags and are small in size. In some cases, there is hypospadias. Thus, individuals with sex reversal have a karyotype of one sex and gonads, both the genitalia and phenotype, of the opposite sex; for example, an individual has an XY karyotype and ovaries and is female (but these cases have not been described in the human species). However, there are XX individuals who have testicles and are men: These are cases of true sex reversal. They often present with signs of androgen deficiency, sometimes with gynecomastia, but may be normal in appearance. There are no somatic abnormalities, and gonadotropins and intelligence are normal. Spermatogonia have occasionally been seen in the ejaculate content, but there appear to be no cases of fertility, so infertility with azoospermia is reported to be universal [61,62].

#### 4.1.5. Klinefelter Syndrome (KS)

In 1942, Klinefelter et al. described nine male patients with a syndrome characterized by “gynecomastia, aspermatogenesis without Leydigism, and an increased excretion of Follicle-Stimulating Hormone FSH” [63]. However, early studies showing positive nuclear Barr chromatin were explained after the discovery by Jacobs et al. [64] that the karyotype of these people is 47,XXY. It is a syndrome that appears in 1/500–1000 children born [65,66], affecting future fertility. Indeed, individuals with KS generally have hypogonadism and infertility, atrophy and hyalinization of the seminiferous tubules, a relative increase in the number of Leydig cells, and small and usually harder testes, although they are sometimes softer depending on the degree of hyalinization. Sometimes, they have spermatogonia or sperm in the semen, but these cases are frequently mosaic. Although 50–75% have enlarged breast parenchyma, only 20% show evidence of gynecomastia. Between individuals there is a higher prevalence of adverse cardiovascular, metabolic, and bone conditions, as well as of various neurocognitive and psychosocial manifestations that could explain the increased morbidity and mortality in KS. Similarly, there is likely a higher prevalence of dental, coagulation, and autoimmune disorders in these patients. Genetic and epigenetic effects due to the supernumerary X chromosome, as well as testosterone deficiency, have been implied in this pathological pattern [67]. The rate of diagnosis remains low relative to prevalence since the majority of patients with KS are diagnosed during adulthood, as infants with KS usually have a normal male phenotype. Currently some cases are diagnosed prenatally, but KS may be suspected in infants with bilateral cryptorchidism and/or micropenis. Later on, delayed puberty, poor testicular development, gynecomastia, high stature, learning disabilities, and psychosocial distress are manifestations that should make KS suspect [67]. All aspects related to this syndrome have been reviewed by a large group of experts appointed by the European Academy of Andrology, generating recently published guidelines on Klinefelter Syndrome [67]. In this consensus guide it is suggested that pediatric and juvenile patients with KS require specific supervision to monitor their development and fertility. So, the recommendations for follow-up are speech therapy; the control of educational problems; social training and psychological assessment in adolescents; and trying to minimize neurodevelopmental dysfunction, that is, verbal deficits, learning difficulties, and problems of behavior. It is considered that these measures are likely to improve the patient’s self-esteem, ensure quality of life, and facilitate their social adaptation. Finally, the aforementioned guidelines consider that the preservation of fertility potential, that is, the cryopreservation of sperm from ejaculate or testicular tissue, is an option that should be considered, which is widely available [67].

Thus, from a fertility point of view, it should be noted that infertility is common and that less than half of the adult men reported by Corona et al. [68] had sperm in their ejaculation, although, in some individuals with apparent azoospermia, there may be residual foci with spermatogenesis. Van Saen et al. [69] reported that there is a progressive loss of spermatogonial stem cells that begins at prepubertal time, with frequent testicular fibrosis by adulthood. A meta-analysis of the presence of spermatogonia in individuals with KS demonstrated their presence in the testes of all fetal/infant samples, in 83% of those obtained from prepubertal patients, and in 40–50% of adolescents/adults [70]. Thus, the options for fertility preservation in individuals with KS depend on the age of the patient. For postpubertal patients, the main goal is to obtain viable sperm that can be stored for future use with artificial reproductive technologies; this, if not collected from ejaculate, would involve a surgical intervention (i.e., testicular wedge) to retrieve sperm using testicular sperm extraction [41]. This technique, in the systematic review and meta-analysis by Corona et al. [68], was shown to successfully retrieve sperm in around 40% of cases. However, interestingly, clinical and biochemical parameters, such as age, testis volume, and hormone status at baseline, were not predictive of success. A total of 29 studies in this meta-analysis reported fertility outcomes using sperm retrieved by testicular sperm extraction resulting in pregnancies and a 43% birth rate. However, for prepubertal boys, the only option is to cryopreserve testicular tissue in order to preserve spermatogonial stem cells that can be used to generate sperm in the future life. However, the underlying abnormalities in the testes and germ cells of individuals with KS may create additional challenges for the use of this approach for fertility preservation. Thus, this approach has recently been questioned [69], and cryopreservation of testicular tissue in prepubertal patients should only be considered as part of an ethically approved research study [41].

#### 4.1.6. Dysgenetic Infertility

As mentioned, some of the infertility situations related to oligoazoospermia in men or to primary ovarian failure in women are minor degrees of gonadal dysgenesis. Portnoi et al. [43] showed that SOX8 is expressed in the somatic cells of the early developing gonad in humans and influences human sex determination. They analyzed SOX8 in a cohort of infertile men (*n* = 274) and two independent cohorts of women with primary ovarian failure (*n* = 153 and *n* = 104). SOX8 mutations were found more frequently in oligozoospermic men (3.5%; *p* < 0.05) and in primary ovarian failure (5.06%; *p* = 4.5 × 10^−5^) compared to fertile/normospermic control populations (0.74%). The mutant proteins identified altered SOX8 biological activity compared to the wild-type protein. These data demonstrate that SOX8 plays an important role in human reproduction and that SOX8 mutations contribute to a spectrum of phenotypes, including male infertility and 46,XX primary ovarian failure. Other cases are related to mutations of the NR5A1 gene that encodes the steroidogenic factor-1 (SF-1) [71], as we will see later.

### 4.2. Anomalies in Sex Determination with Sex Ambiguity (Usually)

These are cases of DSDs due to anomalies in sex determination, that is, due to chromosomal and/or gonadal abnormalities as in group A, but frequently presenting sex ambiguity. This group should include: (XY) testicular dysgenesis (including dysgenetic male Ps and mixed gonadal dysgenesis), TH or ovotesticular (OT) ASD or OT-DSD, and some mutations in the NR5A1 gene (9q33.3)/SF-1). In a study of 408 cases of genital ambiguity followed by a single multidisciplinary team across a 23-year period reported by De Paula et al. [72], there were 95 cases (23.3%) of anomalies of gonadal development, and 19 (4.7%) with complex malformations. In the disorders of gonadal development group, 46,XY partial gonadal dysgenesis, mixed gonadal dysgenesis, and ovotesticular DSDs were more frequent [72].

#### 4.2.1. Testicular Dysgenesis

(1) Dysgenetic male Ps (or XY partial gonadal dysgenesis (PGD)) is characterized by dysgenetic testes on each side, without a gonadal band streak, but with a karyotype and symptoms that are impossible to differentiate from the mixed gonadal dysgenesis that we will analyze later. In fact, Andrade et al. [73], who studied 61 patients with testicular dysgenesis in their service between 1989 and 2013, pointed out that although historically, the terms partial (PGD) and mixed gonadal dysgenesis (MGD) have been used to describe an incomplete testicular differentiation in individuals with the 46,XY or 45,X/46,XY karyotypes, respectively, it is currently unclear as to what extent the clinical features actually differ between these individuals, and they proposed a classification based on the karyotype [73]. The difference between dysgenetic male Ps and sex reversal is also in the karyotype (sex reversal is 46,XX), and that patients with dysgenetic male Ps have Müllerian ducts (derivatives) and present sex ambiguity, whereas individuals with sex reversal are phenotypic males.

(2) Mixed gonadal dysgenesis (MGD) is characterized by the simultaneous existence of a streak gonadal band and testis, either dysgenetic or well developed. Sometimes, a streak band and testicle are seen on both sides. In the literature, MGD is also sometimes referred to as PGD [41], but is actually dysgenetic male Ps. The sexual ambiguity is marked and, in fact, 2/3 of such individuals are raised as women and 30% present with what appears to be Turner syndrome. The karyotype is usually a mosaic of Y, and sometimes XY. Andrade et al. [74] reported a case and again pointed out the difficulties caused by the use of the terms MGD and XY-PGD. The Müllerian derivatives always persist and there is sex ambiguity, varying the phenotype between male and female. MGD is the second most common cause (after congenital adrenal hyperplasia (CAH)) of sex ambiguity. Patients present with an infantile or rudimentary uterus and bilateral tubes; but if the testicle is normal and well differentiated, the tube on that side is usually absent. Wolffian organs are also usually present, and the epididymis is usually on the side of the testis and, less frequently, the vas deferens. Some testes are normal, with spermatogonia, but more frequently, they are dysgenetic; however, normal ovaries are not present, since this would then constitute a TH.

Because they often have Y chromosomes, such individuals are at high risk for gonadoblastoma and dysgerminoma, and therefore, the gonads should be removed before puberty. However, Weidler et al. [75] noted that gonadectomy deprives patients of the benefits of their endogenous hormones and potential fertility, and criticized that patients with MGD have historically undergone gonadectomy before addressing gender identity. Similarly, Kim et al. [76] noted that it is difficult to determine parenting sex and to predict long-term pubertal outcomes in patients with MGD (and ovotesticular DSD), and therefore, long-term follow-up is required to control spontaneous puberty, final sex, and urological and gynecological complications.

Indeed, from the point of view of fertility, in cases with testicular dysgenesis, it has been pointed out that whilst severe oligozoospermia has been reported in a long-term follow-up study of males with PGD [77], for phenotypic males with mild abnormalities of gonadal development or external genitalia (e.g., hypospadias), fertility may be possible [41].

#### 4.2.2. Ovotesticular Disorders or TH (OT-ASD or OT-DSD)

Such disorders are rarer causes of sex ambiguity, but very interesting nonetheless. They are defined by the coexistence of testes and ovaries in the same individual, that is, there is no streak band, but there are all kinds of ovary/testicle combinations, alternating, or ovotestes. The testicle may not produce anti-Müllerian hormone (AMH), or it may only produce AMH on its side. After puberty, the ovary is usually the dominant one, with ovulation in 25% of cases and normal menstruations, whether the individual has the external phenotypic appearance of a man or a woman. Generally, when estrogen is produced, the testes begin to degenerate. The karyotype is variable: 60% XX; 13% XX/XY; 12% XY; 6% XY/XXY (mosaics 28%). Most of these subjects are raised as women. The genitalia are usually ambiguous, and also present other characteristics: in those raised as males, there is gynecomastia, menstrual hematuria, cryptorchidism, hypospadias, hernias, etc.; while in in those raised as women, there may be amenorrhea, endometriosis, ovarian tumors, fibroids, etc., but few women conceive. However, Tiltman et al. [78] described a 52-year-old woman with ovotestes who had had nine deliveries (they observed this during adnexectomy). Tumors such as gonadoblastomas and others (dysgerminoma) have sometimes been reported. Interestingly, in TH XX, the ovary is more frequently found on the left side, and the inappropriate gonad, either ovotestes or testis, is more often on the right side. However, the diagnosis of OT disorders should only be made if has been histologically demonstrated that the ovary and testicular tissue coexist within the same gonad or in opposite gonads. Additionally, the presence of a gonadal stroma (streak gonad) without an appreciable number of oocytes, is not adequate to classify gonadal tissue as ovarian. Moreover, the nature of the internal or external genitalia based on ambisexual appearance is not a sufficient criterion for the classification of an OT disorder either.

From the point of view of fertility, it should be taken into account that the ovarian component of the gonad tends to have a relatively normal histology, whereas the testicular component is more frequently dysgenetic, with limited germ cells and rarely sperm [62]. Gomes et al. [41] pointed out that while spontaneous pregnancies have been reported in 11 women with TH [79], successful paternity has only been described in an infertile male with TH and a 46,XX/46,XY karyotype after extraction of testicular sperm and intracytoplasmic sperm injection (ICSI) [80]. For phenotypic men, the fertility preservation options are similar to those of dysgenetic male Ps. For phenotypic women with OT disorders who cannot achieve a natural pregnancy, the retrieval of oocytes for assisted reproduction may be possible, although this also depends on the presence and anatomy of the Müllerian structures and their potential to carry a pregnancy to term. Recent advances in uterine transplantation have increased the possibilities for people with ASDs who lack Müllerian structures [41,81], but there is no current evidence of its efficacy and safety.

#### 4.2.3. Mutations in the NR5A1 Gene

Fabbri et al. [82] reported that loss-of-function mutations in the NR5A1 gene can lead to a 46,XY PGD phenotype with mild manifestations, dysgenetic testes, and preserved adrenal function. Similarly, Pan et al. [52] reported the clinical characteristics and characteristics of the genetic mutation in an individual with 46,XY DSD caused by a new heterozygous mutation in the NR5A1 gene: c.630delG (p.Y211Tfs*85), presenting the case of a Chinese boy with ambiguous genitalia at birth and normal adrenal glands.

## 5. DSDs due to Anomalies in Sex Differentiation (Hormones and Enzymes)

Among the anomalies in sex differentiation, we must consider the following three groups:

### 5.1. MALE Ps or Male (XY) with Androgen Deficiency (MAD) (by Abnormal Fetal Endocrinology, without Gonadal Abnormality, or Non-Dysgenetic)

Such individuals have a karyotype and gonads that are male (XY and testes), but in terms of the phenotype, their external appearance is female. In other words, these individuals are feminized men, generally due to abnormal fetal endocrinology, abnormalities in steroidogenesis with a lack of androgen formation, or their actions on target organs. Generally, male Ps or MAD can be produced by:
Fetal gonadotropic deficiency: Luteinizing hormone (LH) (or human chorionic gonadotropin -hCG-) that stimulates Leydig cells (Leydig cell hypoplasia), although this is rare. Park et al. [83] described “A Case of Male Pseudohermaphroditism Associated With Elevated LH, Normal FSH, and Low Testosterone Possibly Due to the Secretion of an Abnormal LH,” as a primary defect of the CNS with secretion of an abnormal LH and producing male Ps. In general, if LH is absent, androgens are decreased, the testes do not descend (cryptorchidism), and there is microphalo; however, more frequently, this may lead to gonadotrophin-resistant testes [84].Deficiency of the testicle itself and its secretions; therefore, there may be male Ps by:(a)Embryonic testicular regression, described by Edman et al. in 1977 [85] and later by Coulam in 1979 [86], as testicular regression syndrome (TRS). TRS is attributed to an early regression of the embryonic testicle, and therefore, there are no Müllerian derivatives, unlike in Swyer’s syndrome, in which there is a uterus, but it may depend on the moment of such embryonic regression.(b)Enzyme block in steroidogenesis, and consequently, deficient androgen formation.(c)Poor response to androgens, or defects in androgenic action at the target organ level, which would include complete and incomplete testicular feminization syndromes and 5α-reductase deficiency.(d)Other mild forms of male Ps, or unambiguous sex, including cryptorchidism, hypospadias, and defects in the formation or action of AMH.

Each of these anomalies are analyzed below.

#### 5.1.1. Gonadotrophin-Resistant Testes and Fetal Gonadotropic Deficiency

Individuals with male Ps by agenesis or abnormal differentiation of Leydig cells [87,88] are characterized by poor response to LH/hCG. There may be Leydig cell hypoplasia due to fetal gonadotropic deficiency, but David et al. [89], and Saldanha et al. [90] defined it as a syndrome of gonadotropin resistance possibly due to a luteinizing hormone receptor (LHR) defect. However, these cases are usually grouped under the name of “gonadotropin-resistant testes.” In general, the characteristics of the syndrome include basically female but ambiguous genitalia, cryptorchidism with Leydig cell degeneration (Leydig cell agenesis or hypoplasia), the absence of Müllerian ducts but the presence of vas deferens and epididymis, and high levels of gonadotropins (FSH increases even more after gonadectomy, indicating the presence of inhibin). Gonadotropin resistance can affect both men and women [84,91]. Latronico et al. [92] described cases in two affected families, and DNA sequencing analysis revealed new homozygous mutations of the LHR gene in each kinship, which damaged the function of the LHR, thus preventing the transmission of the hormonal signal to the testes and ovaries of the affected patients. However, normal pubertal feminization in these cases suggests that LH does not play an important role in pubertal development in girls [92]. Indeed, Arnhold et al. [93] later published a study of seven sisters of patients with male Ps due to LH resistance and they concluded that women with LH resistance have spontaneous breast development, primary or secondary amenorrhoea, infertility, elevated serum LH levels and LH/FSH ratios with normal androgen levels, and normal or enlarged cystic ovaries. Therefore, in females, primary and secondary sexual characteristics develop independently of LH action. However, LH stimulation is necessary for normal ovarian steroidogenesis and ovulation [93]. Huhtaniemi and Alevizaki [91] also pointed out that gonadotropin resistance is caused by inactivating mutations in the receptors of the two gonadotropins, LH and FSH, presenting as hypergonadotrophic hypogonadism and infertility/subfertility in both sexes by acting on G protein-coupled receptors. Thus, these conditions are extremely rare, but they must be taken into account in the differential diagnosis of DSDs, hypogonadism, and infertility.

In 46,XY individuals, LHR inactivation causes an alteration in sex differentiation, ranging from male Ps (with total lack of genital masculinization) to mild conditions such as cryptorchidism and hypospadias, depending on the integrity of the receptor inactivation. Simoni et al. [94] also described the polymorphisms of the LH/hCG receptor gene (pituitary LH and placental hCG share the same receptor, i.e., LHCGR) and their association with poorly descended testes and male infertility. In women, the alteration in the phenotype is milder and presents mainly as anovulatory amenorrhea and hypoestrogenization. However, inactivation of FSH receptor causes, in normally masculinized men, testes of a small size and variably reduced spermatogenesis, but not azoospermia or absolute sterility; meanwhile, in women, the phenotype is more severe, with early primary or secondary amenorrhea, arrested follicular maturation, and anovulatory infertility. Incomplete forms with a milder phenotype and a partial response to FSH have also been described; thus, although gonadotropin resistance is a very rare condition, its correct diagnosis is important for the selection of an appropriate treatment [91].

Latronico and Prado-Arnhold [84] also reiterated that the homozygous or heterozygous inactivating compound mutations of LHCGR are associated with a phenotypic spectrum of female or ambiguous external genitalia, including a micropenis, hypergonadotropic hypogonadism, and delayed puberty due to Leydig cell hypoplasia in genetic males. The size of the testicles is slightly reduced and testosterone levels are low in affected men; interestingly though, the clinical phenotypes are closely related to the severity of the mutation. In women, the phenotype is also variable and can range from primary amenorrhea to oligoamenorrhea associated with infertility [84].

As has been pointed out, in individuals with the complete form of the syndrome, the phenotypic aspect, and therefore the breeding sex, is usually female; thus, from a therapeutic point of view, in such cases, pubertal induction with estrogen is required, and then gonadectomy is frequently performed because of the increased risk of gonadal malignancy [41]. However, although males with Leydig cell hypoplasia are considered azoospermic, the case of an individual with an inactivating mutation of the LHR from whom sperm was successfully obtained by extraction of testicular sperm after hCG treatment has been described. The cryopreserved sperm was used for ICSI and resulted in a successful pregnancy and a live child delivery [95].

#### 5.1.2. Embryonic Testicular Regression Syndrome (TRS; Anorchia)

Embryonic testicular regression syndrome (TRS) is a condition that occurs in 46,XY males with phenotypic male genitalia and bilateral absence of the testes (testicular nubbin) [96]. The disorder is therefore due to a regression of the fetal testicle at some point in embryonic or fetal life, and depending on that moment, becomes the clinical aspect. At one extreme would be XY gonadal dysgenesis (Swyer syndrome) with Müllerian derivatives (uterus) and normal female external genitalia; at the other end, they would be infantile males, without Müllerian derivatives (when it occurs after >120d of development), and in between, it would depend on the performance of the AMH, testosterone (T), and dihydrotestosterone (DHT). As a cause of the syndrome, the transcriptional factor DMRT1 has been implicated in human fetal testes, altering the expression of key testicular SD genes. Thus, DMRT1 repression would affect testicular development and maintenance [97,98]. Indeed, McElreavey et al. [99] pointed out that the pathogenic variants of DHX37 are a new cause of an autosomal dominant form of 46,XY DSD, which includes gonadal dysgenesis and TRS, demonstrating that these conditions are part of the same clinical spectrum and suggesting the possibility that some forms of DSD may be a ribosomopathy [99].

Late-affected 46,XY individuals (anorchia) have infantile, but not ambiguous, male external genitalia, and male Wolffian ducts, but there is no presence of Müllerian ducts and they do not have detectable testes. Early testicular function existed (the presence of Wolffian structures and AMH function), but it was not maintained to a certain degree or for long enough for a normal-sized phallus to develop. However, in the case of earlier regression, we have already seen that ambiguity may exist, and the attribution of sex depends on the degree of development of the external genitalia.

Nataraja et al. [100] conducted a systematic review of the syndrome where the primary outcome measure was the incidence or finding of germ cells and the secondary outcome was the incidence of tumors. Twenty-nine pediatric studies with a total of 1455 samples were included in the systematic review. The incidence of tumors was 10.7% (156/1455) and the finding of germ cells was 5.3% (77/1455). Histological analysis, excluding the presence of tumors or germ cells, was consistent with TRS, fibrosis, calcification, or hemosiderin deposits. Intra-abdominal TRS specimens may contain more elements and therefore require excision, but no strong evidence is available to determine whether a TRS specimen in the inguinal or scrotal position requires routine excision [100].

#### 5.1.3. Disorders of Androgen Production (Male Ps or MAD due to Blockage in Steroidogenesis)

In these disorders, the ambiguity (or Ps) is due to the deficient formation of androgens and depends on the level of the stop in steroidogenesis, that is, of the missing enzyme. Therefore, it must be remembered that steroidogenesis and its enzymes, as well as the deficiencies of the enzymes involved in the formation of androgens, can manifest themselves in the adrenal gland and in the testis (see Figure 2).

The first five enzymes whose deficiency can produce male Ps or MAD and in which the Müllerian derivatives will never be present (i.e., uterus and tubes) are the cases of: (1) 20,22-desmolase deficiency or congenital adrenal lipid hyperplasia; (2) deficiency of 3β-ol-dehydrogenase and ∆4-5-isomerase; (3) 17α-hydrogenase deficiency; (4) 17,20-desmolase deficiency; (5) 17β-oxireductase deficiency.

Deficits of 21-hydroxylase and 11-β-hydroxylase cause female Ps, which we discussed, but nevertheless, it should be noted here that in 46,XY males with CAH due to deficiency of 21-hydroxylase, reduced fertility and semen parameters are also reported [101,102,103,104]. Indeed, in men with CAH, the fertility rate is reduced compared to the normal population, the most common cause being the presence of testicular adrenal remnant tumors (TARTs) [102]. Furthermore, the intensification of glucocorticoid therapy does not always lead to tumor regression or improvement of testicular function; neither is the pituitary–gonadal function improved by conservative testicular surgery, despite the successful removal of tumors [105]. Reisch et al. [103] also noted that TARTs (likely reflecting undertreatment) contribute to the high prevalence of subfertility in men with CAH and that inhibin B values are important indicators of fertility. However, long-term glucocorticoid overtreatment also appears to contribute to poor semen quality [103].

The enzymatic disorders that produce male Ps or MAD are discussed below.

(1) Cholesterol 20-22-desmolase or 20-hydroxylase deficiency (steroidogenic acute regulatory (StAR) deficiency; P450 side-chain cleavage (P450scc) deficiency; Cytochrome P450(CYP)11A1). These deficiencies are frequently described as lipid congenital adrenal hyperplasia (LCAH), an autosomal recessive disease described in male children with severe adrenal insufficiency and marked lipid accumulation in both adrenal glands and with testes, but with female external genitalia and a male ductal system. However, XX women with this disorder do not show abnormality in their genitalia, but it is generally incompatible with life. Thus, LCAH (OMIM n.201710) is the most severe form of CAH and is characterized by severe impairment of adrenal and gonadal steroidogenesis due to a defect in the conversion of cholesterol to pregnenolone. Affected children also experience salt loss, but glucocorticoid and mineralocorticoid replacement therapy allows long-term survival. Classic LCAH is relatively common in Japan and Korea, but extremely rare in Caucasian populations [106]. As the StAR protein plays a crucial role in steroidogenesis by accelerating the transport of cholesterol to the inner mitochondrial membrane, where the cytochrome P450scc enzyme is found, mutations of StAR [107,108], as well as of the P450scc enzyme [106,109,110] or the CYP11A1 gene that encodes it (chromosome 15), are causes of this disorder. Recently, Kallali et al. [111] studied three siblings with partial P450scc deficiency, highlighting the importance of an accurate diagnosis in primary adrenal insufficiency to ensure adequate counseling and management, particularly of TARTs. Piya et al. [112] also presented a patient with ambiguous genitalia, a salt-loss crisis within two weeks after birth, and low cortisol levels, with a mutation or absence of exon 1 of StAR.

(2) Deficiency in 3β-hydroxysteroid dehydrogenase/∆4,5-isomerase. The isoenzymes 3β-hydroxysteroid dehydrogenase (3β-HSD)/∆4,5-isomerase (encoded in the HSD3B2 gene) are responsible for the oxidation and isomerization of the ∆5-3β-hydroxysteroid precursors in ∆4-ketosteroids, thus catalyzing a step essential in the formation of all classes of active steroid hormones. In humans, the expression of the type 1 isoenzyme accounts for the 3β-HSD activity found in the placenta and peripheral tissues, whereas the type 2 3β-HSD isozyme is predominantly expressed in the adrenal gland, ovaries, and testes, and its deficiency is responsible for a rare form (less than 1%) of CAH [113]. 3β-HSD is essential for the formation of progesterone (P), 17-OHP, and androstenedione (∆4A), which are the precursor hormones for aldosterone, cortisol, and T, respectively. Thus, salt loss can occur in both men and women, but in men, 3β-HSD deficiency is characterized by incomplete virilization of the external genitalia due to impaired androgen biosynthesis in the testes, while women exhibit mild virilization or normal external genitalia [114]. Most patients with this disorder die in infancy. However, a few individuals have survived to puberty, with persisting adrenal insufficiency. Bahíllo-Curieses et al. [115] reported a partial deficiency of type 2 3ß-HSD, diagnosing a new mutation after positive screening in newborns for 21-hydroxylase deficiency.

(3) 17α-hydroxylase deficiency (CYP17A1 gene). When there is a deficiency in the enzyme 17α-hydroxylase, there is also a deficiency of the C21 steroids derived from pregnenolone and progesterone (P): 17α-hydroxy-pregnenolone and 17α-hydroxy-progesterone (17α-OHP), as well as the C19-derived steroids. 17α-hydroxylase (and 17,20-lyase) deficiency is one of the two hypertensive forms of CAH that is inherited as an autosomal recessive trait. Although rare, it likely exists twice as often as 11β-hydroxylase deficiency. However, it is noted that it is one of the less common forms of CAH, corresponding to approximately 1% of cases, with an estimated annual incidence of 1 in 50,000 newborns [116], and it is described more frequently in women. The deficiency of this enzyme results in the absence of the formation of sex hormones in both the adrenal glands and the gonads, causing hypogonadism and male Ps. The elevated and suppressible glucocorticoid levels of the zona fasciculata (ZF) 17-deoxysteroids, namely, 11-deoxycorticosterone (DOC) and corticosterone, as well as their 18-hydroxylated products -18-OHDOC and 18-OHB (in addition to 19-nor-DOC)- are responsible for hypertension, hypokalemia, and renin and aldosterone suppression [117]. Therefore, males with this disorder have male Ps with deficiency in the development of secondary sexual characteristics; meanwhile, in women with this disorder, as described by Kater and Biglieri [117], the secretory rates of steroids not dependent on 17α-hydroxylated precursors are elevated. These include P, DOC, and corticosterone. Although the presence of these steroids allows the survival of patients, hypertension and hypokalemic alkalosis are the main problems encountered. As would be expected, in women, there is practically zero estrogen production with consequent sexual infantilism; thus, the treatment consists of replacement doses of glucocorticoid hormones and supplemental estrogen therapy in young adult patient [117]. Dhir et al. [118] described the functional characterization of four mutations (i.e., A174E, V178D, R440C, L465P) in the CYP17A1 gene, resulting in steroid 17α-hydroxylase deficiency. Two patients with 46,XY DSD (MAD) presented, at the ages of 5.5 and 8.8 years, respectively, with tall stature and hypertension. Mutation analysis revealed heterozygous compound mutations of CYP17A1 (A174E/K388X; V178D/R440C). The third patient (46,XX) presented with primary amenorrhea and hypertension at 15 years of age, and she was homozygous for the new L465P mutation. More recently, Espinosa-Herrera et al. [116] reported a case of CAH due to 17α-hydroxylase deficiency in two phenotypically female 46,XX sisters, both with primary amenorrhea, a hypergonadotropic hypogonadism profile, the absence of secondary sexual characteristics, and osteoporosis. High blood pressure was also present in the older sister. There was a decrease in cortisol and dehydro-epi-androsterone sulfate (DHEA-S), an increase in adrenocorticotropic hormone (ACTH), normal 17-OHP levels, but very high levels of DOC, and the CT scan showed bilateral adrenal hyperplasia in both sisters. Consanguinity was evident in their ancestors.

(4) 17,20-desmolase or 17,20-lyase deficiency (cytochrome P450c17, P450 17A1, CYP17A1). Unlike the three enzymatic deficits described above that affect adrenal corticosteroid and androgen synthesis, isolated 17,20-desmolase or 17,20-lyase deficiency affect only androgen formation (and subsequent estrogens), leading to sexual infantilism [119]. However, as previously noted, CYP17A1 catalyzes 17α-hydroxylase and 17,20-lyase reactions, regulating the steroid hormones produced by the adrenal glands and gonads, thus individuals with clinically combined 17α-hydroxylase/17,20-lyase deficiency present with hypertension, hypokalemia, primary amenorrhea, and sexual infantilism. However, some mutations selectively impair 17,20-lyase activity, and others that affect cytochrome P450-oxidoreductase and cytochrome b5 cofactor proteins also selectively alter 17,20-lyase activity, leading to a defect in the synthesis of sex steroids that impairs fertility in male and female patients when the deficiency is severe. Zachmann et al. [120] described steroid 17,20-desmolase deficiency as a new cause of male Ps or MAD, showing that family members with this disorder can also have ambiguous external genitalia, inguinal or intra-abdominal testicles, and severe hypospadias, with a male-type urethra and the development of male ducts [120,121,122,123]. If the diagnosis is made in infancy, T treatment will result in adequate penis growth and the development of male secondary sexual characteristics [120,121,122,123]. Miller reviewed and updated the syndrome of 17,20-lyase deficiency [124].

(5) 17β-hydroxysteroid dehydrogenase deficiency. 17β-hydroxysteroid dehydrogenase type 3 (17βHSD-3 or 17β-oxireductase) is the enzyme that catalyzes the conversion of androstenedione (Δ4A) to T in the testes of the developing fetus, thus playing a crucial role in the differentiation of the gonads and in the establishment of male phenotypic sex. This enzyme (17β-oxireductase) is also required for the conversion of estrone to estradiol (E2). 17β-HSD-3 deficiency is an autosomal recessive disorder with decreased T synthesis and, consequently, subandrogenization; hence, 46,XY patients with this deficiency are often assigned a female sex at birth, although they have a high potential for virilization at puberty [125]. The diagnosis can easily be lost in early childhood, as the clinical presentation can be subtle [126]. Any girl with an inguinal hernia, mild clitoromegaly, a single urethral opening, or a urogenital sinus should raise suspicions [127]. If not diagnosed early, patients develop severe virilization and primary amenorrhea in adolescence and may experience a role shift from female to male. A low T/Δ4A determination in a baseline or hCG-stimulated test suggests a 17βHSD-3 deficiency, although the diagnosis must be confirmed with molecular genetic studies.

Thus, 17β-HSD-3 deficiency syndrome is a rare form of unfamiliar MAD, an autosomal recessive disorder in 46,XY males, associated with pubertal gynecomastia and virilism, which is due to pathogenic variants in the gene HSD17B3. The mutated genes encode an abnormal enzyme with absent or reduced ability to convert Δ4A to T in the fetal testicle, and affected individuals are generally raised as women with diagnosis made at puberty, when they show virilization. However, it is perfectly possible that many patients are classified as having incomplete form of testicular feminization or that they really correspond to this deficiency, which will be described later. However, in contrast to classic testicular feminization, these 17β-HSD-deficient patients have a defect in testicular T formation, while androgen sensitivity of the target tissues is normal. Because of the blockage in the formation of T and E2, there are increased rates of Δ4A and estrone secretion by the testes. Additionally, the conversion of Δ4A to T at extratesticular sites occurs less efficiently than within the testis, but increased amounts of Δ4A eventually occur, resulting in amounts of T sufficient to cause virilization. Failure of spermatogenesis and small tubules may be the result of intratesticular androgen deficiency. Moreover, a lack of intrauterine fetal virilization may be a result of placental aromatization of Δ4A, thus obviating the extratesticular conversion to T. LH levels, which are increased after puberty, stimulate an increase in Δ4A and estrone so that the peripheral conversion to T and E2 could cause simultaneous virilization and gynecomastia. As Goebelsmann et al. [128] already pointed out, because feminization in this syndrome is the result of increased amounts of estrogen originating directly or indirectly from the testicle, perhaps the name testicular feminization syndrome would be a more appropriate term for 17β-HSD deficiency than for the entity for which it was initially used and described.

#### 5.1.4. Male Ps or MAD Due to Defects in Androgenic Action

This category includes the so-called androgen insensitivity syndromes (AISs), i.e., androgens formed in patients that do not act on the target organ. At least four hereditary syndromes should be noted in relation to androgenic action defects: (1) the complete form of testicular feminization or Morris syndrome (CAIS); (2) incomplete or partial testicular feminization (PAIS); (3) Reifenstein syndrome (incomplete male X-linked recessive Ps); (4) autosomal recessive male Ps due to 5α-reductase deficiency. However, as will be seen below, there are more disorders that are not expressed as Ps. It must be remembered that the high affinity receptor for androgens, responsible for the actions of T and DHT, is already present in female and male embryos; thus, the differences in phenotypic development in both sexes are due solely to the differences in the hormones produced by the gonads during the critical period of embryonic development. A mutation of the androgen receptor (AR) gene on the X chromosome (Xq11.2q12) is the main cause of AISs. These syndromes are disorders in which the involution of the Müllerian duct and the synthesis of T are normal, but in which, as a result of a genetic mutation involving a single gene, affected persons are resistant to androgens by default of some of the mechanisms of action of these androgens. These alterations were originally recognized in patients with disorders severe enough to produce male Ps, but later, men with less severe disorders and in whom infertility was observed as the only clinical manifestation of resistance to androgens were identified.

The molecular defects responsible for this resistance to androgens may reside in one of the three main mechanisms on which the action of androgens is based: Alterations in 5α-reductase, defects in the AR, or abnormalities in the subsequent phases of androgen action (receptor-positive resistance), as described below.

(1) 5α-reductase deficiency 

5α-reductase deficiency (5α-RD) was originally called pseudovaginal perineoscrotal hypospadias, and it is currently known to be due to the deficient conversion of T to DHT due to a mutation in the SRD5A2 gene located on the short arm of chromosome 2. The typical patient is a 46,XY male with normal levels of T and Wolffian structures that are also normal, but who has a predominantly female development of the urogenital sinus and external genitalia, so that his external phenotype is female. At birth, there is usually a certain increase in the size of the phallus. The Wolffian ducts end in a pseudovagina in a cul-de-sac that varies in size and opens into the urogenital sinus or the perineum.

Affected individuals are usually considered girls at birth, but at the time of puberty, they present a variable virility of the external genitalia and develop male axillary and pubic hair. The breasts continue to present an infantile structure (that is, in males, there is no gynecomastia) and there is less facial and body hair, as well as lower degrees of temporal alopecia than in their unaffected male brothers. The testes are well-developed, but spermatogenesis is usually incomplete and prostate tissue is not detected. The degree of masculinization during sexual maturation can be surprising, and some of these individuals who are not treated until after puberty develop an inversion in their patterns of sexual behavior, i.e., those who were considered as girls begin to function as boys. Thus, this syndrome differs from the incomplete forms of testicular feminization that we will discuss later, in that virilization occurs at puberty and that female breasts do not develop. The studies by Imperato-McGinley et al. [129,130] and Walsh et al. [131] demonstrated that structures derived from the Wolffian ducts (i.e., seminal vesicles, ejaculatory ducts, epididymis, and vas deferens) are male, while those structures that they develop from the urogenital sinus and the anlage of the external genitalia are female. Thus, male differentiation failure is limited to the urogenital sinus and rims, the phallus, and the urogenital folds, since these latter structures are under the influence of DHT that cannot be adequately formed in patients with 5α-RD. Low (but normal male) rates of estrogen production may explain the lack of development of gynecomastia. In the 17 families of Santo Domingo studied by Imperato-McGinley et al. [130], affected patients were 46,XY males who had bilateral labioscrotal or inguinal testes. Of greatest interest was that the psychosexual orientation was male post-puberty, although all these patients in their isolated community had been considered female prior to puberty. Recent studies in India [132] have also shown that the most common cause of male Ps is 5α-RD2: In 29 patients with 46,XY DSD, 17 (58.6%) had 5α-RD2 [131]. 

In summary, 5α-RD2 is one of the most important causes of ambiguous genitalia in children. The phenotype can vary from underdeveloped male genitalia to a complete female phenotype [133]. It is a familiar form of male Ps (inherited autosomal recessive disease) and children are frequently registered as female at birth due to the persistence of a urogenital sinus, although virilization is observed later in puberty, resulting in fertile men. This virilization at puberty (the penis syndrome at 12) and the absence of gynecomastia is what differentiates such individuals from incomplete testicular feminization. DHT treatment contributes to the development of the penis and the prostate, which are favorable for the potential fertility of adults with 5α-RD2. However, delayed penile growth and the risk of prostate overgrowth may complicate post-pubertal management with DHT for these 5α-RD2-deficient men [134].

(2) Androgen receptor function disorders

As already noted, a mutation of the AR gene on the X chromosome (Xq11.2q12) is the main cause of AIS. However, disorders in AR functioning can produce a wide spectrum of phenotypic alterations, such as testicular feminization, Reifenstein syndrome, and infertile male syndrome. Despite the differences in their clinical manifestations, these disorders are similar at the endocrinological and pathophysiological level, as well as in their transmission mechanism, as described below.

(2,1) Complete form of testicular feminization—Morris syndrome, or CAIS. Total androgen insensitivity was first described in detail by Morris at Yale, who coined the term “testicular feminization.” Morris [135] clinically described 82 female patients with testes but a female phenotype and for this reason, he named the syndrome testicular feminization. Patients typically seek medical assistance due to an inguinal hernia or for primary amenorrhea. The phenotype is that of a normal woman, except that the axillary and pubic hair is diminished or even absent. The development of the breasts and the distribution of body fat are female; the external genitalia are unequivocally female and the clitoris is normal. The vagina is short and ends in a cul-de-sac, while the internal genitalia do not develop (Wolffian or Müllerian structures cannot be identified), except for the testicles, which may be located in the abdomen, in the inguinal channel, or on the labia majora. Histological examination of the testes reveals incomplete or absent spermatogenesis and normal or hyperplastic Leydig cells. Often, there is a history of family members with a similar affectation, but approximately one third of patients have no antecedents; thus, it is assumed that they represent new mutations of this disorder linked to the X chromosome. Estimates of the incidence of testicular feminization range from one case in every 20,000 male newborns to one case in every 64,000. It is the third most common cause of primary amenorrhea, after gonadal dysgenesis and congenital absence of the vagina.

(2,2) Incomplete form of testicular feminization, or PAIS. Approximately 10% of patients with testicular feminization have an incomplete or partial form of this disorder (i.e., PAIS). Such forms are similar to the complete form, except that there is partial fusion of the labioscrotal folds, normal pubic hair, and partial virilization (development of a slight clitoromegaly), as well as feminization and growth of the breasts at the time of puberty. The vagina is short and ends in a cul-de-sac. Unlike the complete form of the disorder, underdeveloped Wolffian structures can be identified. A family history, where known, is also usually compatible with an X-linked recessive disorder. Clinically, these patients mimic the complete form of testicular feminization, with a male genotype (46,XY), inguinal or intra-abdominal bilateral testicles, and breast development at puberty. However, in contrast to the full form, there may be phallic enlargement at birth, and then at puberty, some degrees of virilization may occur.

(2,3) Reifenstein syndrome. The term “Reifenstein syndrome” is currently applied to a certain variety of disorders also linked to the X chromosome, which also occurs with PAIS, being initially described by Reifenstein [136] and also studied and defined by Gilbert Dreyfus et al. [137] and Rosewater et al. [138] as familial gynecomastia. Originally, it was assumed that each phenotype constituted a different entity, being designated by a different eponym. However, in several family trees, affected members have different phenotypes; thus, it is currently thought that all of these syndromes represent mutations of a single gene. The predominant phenotype is male and the spectrum of defective virilization ranges from the appearance of gynecomastia and azoospermia to hypospadias with the presence of a pseudovagina. However, the usual phenotype is male with perineoscrotal hypospadias and gynecomastia. The axillary and pubic hair is normal, but the facial and thoracic hair is minimal. Cryptorchidism with small testes in which spermatogenesis is incomplete is common, as is hypoplasia of the Wolffian derivatives. The psychological development and sexual behavior of most of these patients is male.

(2,4) Infertile male syndrome. This syndrome is the latest type of resistance to androgens that has been recognized and that, contrary to the previous disorders, does not constitute a form of male Ps; individuals with this syndrome only present infertility. Although some of the affected men in families with Reifenstein syndrome present azoospermia as the only manifestation of the alteration in the receptor, it was not previously been thought that resistance to androgens could be a cause of infertility in men with a negative family history. However, the study of normal men with infertility due to azoospermia or severe oligospermia suggests that there may be an alteration in the receptor in a significant number of these patients.

In summary, with respect to AISs, patients with Morris syndrome are normal-looking women, well-feminized, but without hair and with inguinal testes; they have no Müllerian derivatives and the origin of the syndrome is related to a recessive gene linked to the only X chromosome. The inguinal testes produce T, but since there are no receptors for it, it does not act; therefore, all of the T is converted to E2 and the individual thus becomes well-feminized. For this reason, the breasts grow, and yet there is no pubic or axillary hair. Since there are also no hypothalamic–pituitary receptors for T or E2, the gonadotropins FSH and LH increase and the Leydig cells produce more androgens and therefore more estrogens. Such individuals usually seek treatment for primary amenorrhea or tumors in the inguinal gonads, which must be removed. However, some men with partial androgen insensitivity respond to androgen therapy with increased virilization, but there is no test or determination to predict the likely response and thus this therapy should only be used to reinforce sex assignment as male in cases of partial or incomplete Ps. Therefore, sex attribution can be problematic when there are ambiguous genitalia due to partial receptor response. In cases where female sex is attributed, early gonadectomy should be performed to avoid neoplasia. In Reifenstein syndrome, the phallus may be long enough to attribute male sex at birth, despite perineal hypospadias. However, after puberty, poor androgen receptor function becomes apparent. Thus, the receptor function is inadequate to respond to the androgen peak of puberty, and without the effect of such androgens, estrogen activity predominates, producing feminization with gynecomastia. Such individuals are infertile and do not respond to exogenous androgens. Furthermore, the karyotype is 46,XY normal male, which distinguishes this syndrome from other feminization syndromes of puberty in phenotypic males (such as Klinefelter syndrome).

(3) Receptor-positive resistance

In the analysis of a family in which testicular feminization was due to androgen resistance, but in which the AR appeared to be qualitatively and functionally normal, Amrhein et al. [139] indicated that androgen resistance could be due to the existence of alterations in their action at the post-receptor level. However, as better techniques for the characterization of AR have been developed (monoclonal antibodies and genetic tests), a greater number of cases of receptor-positive resistance have shifted toward the category of qualitatively anomalous receptors. In other cases, a disorder in the action of androgens can actually involve events distal to the receptor, such as abnormalities in the binding of the receptor to DNA.

#### 5.1.5. Other Mild or Unambiguous Sex Forms of MAD.

These include the aforementioned “infertile male” syndrome, congenital cryptorchidism, isolated hypospadias, and defects in the formation or action of the AMH.

(1) The “infertile male syndrome” has already been discussed above.

(2) Congenital cryptorchidism (undescended testis) is one of the most common congenital urogenital malformations in children and was recently reviewed by Rodprasert et al. [140], who reported that the prevalence at birth among children born with normal weight ranges between 1.8% and 8.4%. This condition is associated with a future risk of poor semen quality and an increased risk of developing testicular germ cell tumors and is related to the RXFP2 gene (13q13.1), but as already mentioned, androgens and INSL3 (at 19p13.11) also play a role in the process of testicular descent. This explains the higher prevalence of cryptorchidism among children with diseases associated with a congenitally decreased secretion or action of androgens, such as patients with congenital hypogonadism and AIS. Patients with CAIS were shown to exhibit a higher rate of cryptorchidism than PAIS in the 39 cases of AIS studied by Liu et al. [141]. There is also some evidence that supports cryptorchidism being associated with decreased testicular hormone production later in life, and it has been shown to affect mainly Sertoli cell function in the long term, but it can also affect Leydig cells. The loss of germ cells that occurs in cryptorchidism is proportional to the duration, and therefore, early orchidopexy to descend the testicle into the scrotum is the standard treatment. However, evidence regarding the benefits of early orchidopexy for testicular endocrine function is controversial, and also hormonal treatments that use hCG or Gonadotropin-Releasing Hormone (GnRH) to induce testicular descent have low success rates. More research is needed to evaluate the effects of hormonal treatments on future male reproductive health [140].

(3) Hypospadias. Hypospadias is a relatively common abnormality in children, affecting approximately 1/150 newborn males [142], and is associated with the MAMLD1 (Xp28) and ATF3 (1p32.3) genes for 46,XY-isolated hypospadias [143]. Androgens influence the masculinization of the genital tubercle between 8 and 12 weeks of gestation, and control tubularization of the urethra from the perineum to the tip of the glans penis. If the process is interrupted, hypospadias occurs, with variable proximal urethral meatus, failed ventral preputial development resulting in a dorsal hood, and a discrepancy in the ventral/dorsal length of the penis that results in a ventral curving of the penis, known as chordee [142]. The external urethral meatus can present various degrees of mispositioning, can be associated with a pathological penile curvature, and, depending on the location of the defect, patients may have an additional genitourinary malformation [144]. Surgery is recommended between 6 and 18 months [142].

(4) Defects in the formation or action of the AMH: a lack of regression of the Müllerian ducts. Such defects are currently described as persistent Müllerian duct syndrome (PMDS) and develop due to AMH deficiency or target organ insensitivity to AMH in individuals with a 46,XY karyotype. Thus, PMDS must be caused by mutations in the AMH gene or its receptor, and the mutation is inherited as an autosomal or X-linked recessive trait. PMDS is characterized by a normal male phenotype of the external genitalia, associated with the persistence of Müllerian structures [145]. It is also known as infantile uterine hernia syndrome, and patients are sometimes seen with bilateral cryptorchidism, inguinal hernias, and external genitalia, that, on the other hand, appear normal. Later, when herniorrhaphy is performed, the uterus and Fallopian tubes can be found in the inguinal canal. Naturally, the ovaries are not present, since the gonads are testes, and those derived from the Wolffian ducts are also of the normal male type. Fertility is generally preserved, although there is often no sure evidence of fertility. If a dysgenetic gonad or tumor is observed on the operation, instead of bilateral testicles, then this would indicate mixed gonadal dysgenesis.

### 5.2. FEMALE Ps or Female (XX) with Androgen Excess (FAE) 

Patients with female Ps or FAE have female gonads (ovaries) and a 46,XX karyotype, but present varying degrees of external genital virilization. There is no karyotypic abnormality in this disorder, no abnormality of the ovary or its functional capacity, and no abnormality of internal genital development. Thus, the uterus, tubes, and ovaries are normal, and in contrast to most of the other abnormalities of sex development discussed earlier, reproductive function is often possible after appropriate treatment. Female Ps or FAE is generally the result of an enzyme abnormality that affects adrenal steroid biosynthesis and leads to congenital adrenal hyperplasia (CAH). Less frequently, it can occur as a result of a tumor or the maternal ingestion of preparations containing steroids with androgenic properties. Thus, the most frequent cause of FAE, and in turn, the most frequent intersex or ambiguous state, is CAH. The degree of virilization depends on the moment of action of the androgenic effect (before or after 12 weeks), the most important effects being those that affect the external genitalia.

If the noxa acts before 12 weeks, the androgenic stimulus will produce severe ambiguity, with a penile urethra and the external appearance of a male, except that there are no testes, but the internal genitalia and upper vagina are normal.In less severe cases, there is scrotal hypospadias, with the sinus and vagina communicating with the urethral opening.If androgen acts after 12 weeks, then the external genitalia are normal and there is only an enlargement of the clitoris as a sign of virilization.

These pathologies are described in more detail below.

#### 5.2.1. Congenital Adrenal Hyperplasia in Female/Adrenogenital Syndrome (AGS)

Cortisol and aldosterone are key to the hormonal products secreted by the adrenal cortex. Clinically, the manifestations of steroid production abnormalities result both from the degree of cortisol and aldosterone deficiency, as well as from the biological activity of the steroid precursors produced in excess as a result of the affected enzyme blockage. ACTH acts on the passage between cholesterol and pregnenolone to stimulate adrenal steroidogenesis, and ACTH production is regulated by the amount of cortisol present in the circulation through a negative feedback mechanism. Cortisol can act directly on the pituitary gland, indirectly on the corticotropic releasing factor of the hypothalamus, or a combination of both. Thus, when the amount of cortisol is decreased, the production of ACTH increases. Indeed, in all forms of CAH in which genital tract abnormalities occur, there is a decreased ability to produce cortisol in the face of increased ACTH overproduction. Additionally, with the excessive secretion of pituitary ACTH results an increased production of steroids in the previous steps of the steroid biosynthesis process in which the enzyme blockade is located. That is, there is an accumulation of all of the steroids that precede the blockage, which are present in the circulation in increased amounts. Furthermore, the lowered cortisol stimulates ACTH production, and the decreased formation of mineralocorticoids increases renin–angiotensin production.

These enzyme deficiencies are transmitted with autosomal recessive characteristics and all can occur in both men and women. When a male is affected and there is an abnormal development of the genital tract, a male Ps or MAD is originated, as previously described. When a woman is affected and masculinization of the external genitalia occurs in varying degrees, female Ps or FAE manifests. The most common enzymatic defects are those affecting 21-hydroxylase (P450c21), 11β-hydroxylase (P450c11), and 3β-HSD. Very rarely, the blockage of cortisol synthesis is caused by a defect in P450c17. However, other forms of CAH have been identified that can affect the development of the female genital tract; three of them (i.e., deficiency of 20-hydroxylase or 20,22-desmolase (P450scc), 3β-HSD/∆4,5-isomerase, and 17α-hydroxylase) have already been analyzed in the section on MAD. The defects are similar to what occurs in men in terms of adrenal failure; however, in girls, only genital infantilism is produced due to the lack of estrogens, or there may be some virilization in the second deficiency mentioned. There is also usually no ambiguity or female Ps in 17α-hydroxylase deficiency, but this will be analyzed in women later, as well as the deficiency of the cytochrome P450 oxidoreductase (PORD) enzyme that is responsible for the activity of several other P450 enzymes, including CYP21A2, CYP17A1, and CYP19A1. However, failures in 21-hydroxylase and 11β-hydroxylase are the most important cases of adrenogenital syndrome (AGS) and female Ps or FAE. An additional rare form is 18-hydroxysteroid dehydrogenase deficiency, but this leads to deficiency in aldosterone production and does not affect the development of the genital tract. We will therefore analyze, in more detail, CYP17A1 deficits in women (generally without sexual ambiguity), PORD, and above all, failures in 21- and 11β-hydroxylase.

(1) CYP17A1 deficiency (17α-hydroxylase) in females. As mentioned previously, CYP17A1 catalyzes the 17α-hydroxylase and 17,20-lyase reactions, controlling the steroid hormones produced by the adrenal glands and gonads. Its mutations are extremely rare and cause combined 17α-hydroxylase/17,20-lyase deficiency, so patients clinically present with hypertension, hypokalemia, primary amenorrhea, and sexual infantilism; thus, the defect in sex steroid synthesis impairs fertility in both male and female patients when the deficiency is severe [146]. The decreased enzymatic activity from CYP17A1 is associated with infertility due to hypergonadotropic hypogonadism. In females, folliculogenesis is arrested, and in males, there is testicular atrophy with interstitial cell hyperplasia and arrested spermatogenesis. Although generally anovulatory, there have been some case reports of females with 17α-hydroxylase deficiency who underwent spontaneous menarche with cyclic menses.

(2) Cytochrome P450-oxidoreductase deficiency (PORD). PORD is a disorder of steroidogenesis that causes DSDs, adrenal insufficiency, and skeletal malformations [147]. It is an autosomal recessive disease typically diagnosed in neonates and children with ambiguous genitalia and/or skeletal abnormalities. It is responsible for the decreased activity of several P450 enzymes, including CYP21A2, CYP17A1, and CYP19A1, that are involved in adrenal and/or gonadal steroidogenesis [148]. In a study by Yatsuga et al. [147] on the clinical characteristics of PORD in Japan, it is shown that in many cases, PORD can be diagnosed at <3 months of age. Hydrocortisone, as the primary treatment during infancy, can be used daily or just in stressful situations; however, because patients generally have mild-to-moderate adrenal insufficiency, some might be able to avoid the treatment. Parween et al. [149] pointed out that mutations in cytochrome P450 oxidoreductase (POR) can cause different forms of CAH, and they reported a novel R550W mutation in POR identified in a 46,XX patient with signs of aromatase deficiency as virilization.

(3) 21-Hydroxylase enzyme deficiency (P450c21)

Although this enzyme deficiency affects only the adrenal gland, 21-hydroxylase blockade is the most common form of CAH (95% of cases), as well as the most common cause of sex ambiguity and the most common endocrine cause of neonatal death. In the severe and uncompensated blockage, virilization is accompanied by salt loss and shock. Even in the less severe variants, with sufficient cortisol production, virilization caused by excess androgens can manifest already in utero. Three different clinical presentations are recognized according to severity: salt wasting, simple virilizing, and nonclassic (previously called late-onset or acquired adrenal hyperplasia). The first and second are associated with female Ps at birth, while the third usually manifests in adolescence or later, and produces hirsutism, menstrual irregularities, and infertility.

As patients with this deficiency cannot form cortisol in normal amounts, there is a compensatory increase in ACTH, which is what leads to adrenal gland hyperplasia. This excess ACTH leads to increased steroid production to the point corresponding to enzyme blockage, with a resulting increase in the amount of 17-OHP produced. Therefore, the pathognomonic finding is an increase in circulating 17-OHP and increased excretion of its urinary metabolite, pregnantriol. On the other hand, increased ACTH stimulates the production of adrenal androgens, DHEA, and perhaps Δ4A and T, since the enzyme deficiency does not affect this part of the steroid biosynthesis process. Additionally, if the enzyme defect is severe, there will be a marked decrease in aldosterone production as well.

In medium or moderate forms of the disorder, the primary manifestations are those due to excessive androgen production. The excess of androgen is first expressed in the female fetus in utero, leading to masculinization of the external genitalia to varying degrees. Indeed, the degree of masculinization is, as we have already seen, probably dependent on the stage of intrauterine development of the external genitalia at the time the excess androgen action. Moreover, it is probably also related to the severity of the enzyme defect. The most common is that at birth, there is an enlargement of the phallus, so that it is larger than the normal female, but smaller than the normal male. Instead of a separate urethral and vaginal opening, there is a urogenital sinus that is often covered by tissue resulting from the posterior fusion of the labioscrotal ridges. Thus, varying degrees of external genital abnormality can be found, ranging from an apparently normal perineum to a phallic urethra. If the disorder is not recognized or is not treated appropriately, the increased androgens will cause excessive growth in terms of the length of long bones and heterosexual precocity. Therefore, in childhood, the patient is typically taller than those of the same age, with developed pubic and axillary hair, and phallic hypertrophy. Additionally, in the presence of a continued excess of androgens, the epiphyses fuse prematurely, so that upon reaching adulthood, the patient is smaller than those corresponding to her age; she will also have a masculinized body habit. Furthermore, due to the excess of androgens, pituitary gonadotropins are suppressed, and therefore, ovarian function is not stimulated and the patient will not experience female secondary sexual development or menstrual cycles.

In the most severe forms of the enzyme deficiency, there is a frank deficiency of DOC and aldosterone due to deficient 21-hydroxylation of progesterone, in addition to excess androgens. A lack of mineral corticosteroids leads to “salt-wasting” forms of the disorder, with severe hyponatremia, hyperkalemia, and marked dehydration. Thus, if not recognized and treated properly, the disorder can be fatal.

In short, in the 21-hydroxylase deficit, there is: (1) adrenal failure; (2) an increase in androgens due to steroidogenetic deviation, leading to growth of a phallus and to fusing of the labioscrotal rims; (3) increased androgens that slow down the ovary; (4) a lack of mineralocorticosteroids that leads to salt loss alterations.

(4) 11-β-hydroxylase (P450c11) deficiency 

Clinically, as a result of 11-β-hydroxylase deficiency, virilization and hypertension are observed, as well as marked hyperpigmentation of the external genitalia, common in CAH. Virilization is the result of adrenal androgen overproduction and can be manifested by rapid initial somatic growth or early closure of the epiphyses, phallic enlargement, premature appearance of pubic and axillary hair, mammary hypoplasia, and amenorrhea. Hypertension results from the overproduction of DOC, after which the stop is found. Treatment of this 11-β-hydroxylase deficiency consists of the administration of exogenous glucocorticoids in amounts sufficient to inhibit ACTH secretion, thereby suppressing adrenal androgen and DOC secretion. However, as with other forms of CAH, excessive glucocorticoid administration can inhibit linear growth. Furthermore, although aldosterone production is decreased, DOC production compensates for it. Therefore, mineralocorticoid substitution is not necessary in this form of CAH.

Fertility in FAE due to CAH: XX women with CAH due to 21-hydroxylase deficiency may be fertile [39,41], but these women are also reported to have lower pregnancy rates compared to age-matched controls [150]. Live birth rates of 33%-50% have been reported in “simple virilizing” forms of CAH, but the probability of fertility is significantly lower in severe “salt-wasting” forms of CAH in whom live birth rates have been reported to be between 0% and 10% [151]. In nonclassical forms of CAH, due to partial enzyme deficiencies, live birth rates are higher (63%-90%) than in classic CAH and are similar to age-matched controls [151]. In other, rarer forms of CAH (e.g., female 46,XX) in which gonadal steroid production may also be affected, fertility is rarely described, except in some cases of reports of spontaneous pregnancy; for example, in an individual with a severe 11β-hydroxylase deficiency [152] and in successful in vitro fertilization (IVF) in individuals with 17α-hydroxylase deficiency [153,154,155], as well as in another individual with congenital LCAH [156]. In CAH (46XX) women, infertility can occur as a result of anovulation, menstrual irregularities, thickening of the cervical mucus, and anatomical factors [39], but there are also other important psychosocial factors that affect fertility in people with CAH and that may include the lack of a stable relationship or a smaller number of those seeking pregnancy compared to the general population [150]. In general, the data indicate that the probability of fertility is associated with the severity and control of the disease, and therefore, the basis of treatment is to optimize androgen levels with adequate steroid therapy [39,41].

#### 5.2.2. Hormone Therapy (Iatrogenic FAE)

It should not be forgotten that androgens administered to the mother, nor-derived gestagens, or a maternal virilizing tumor can also be the cause of female Ps. Masculinization in these cases is usually shown by an increase in the clitoris and varying degrees of labioscrotal fusion, depending on the time of gestation and the time of therapy. However, this virilization does not progress with age (as it does with CAH), and female characteristics appear at puberty. The only indicated treatment may be surgical correction of the abnormal external genitalia. 

#### 5.2.3. Maternal Virilizing Tumor

Virilizing tumors can be primary or metastatic in the ovary, and luteoma gravidarum is the most common type. Elevated hCG levels lead to theca-luteal hyperplasia, sometimes with signs of masculinization in the mother in the third trimester. However, maternal androgen-secreting tumors are rarely seen in pregnancy.

#### 5.2.4. Aromatase Deficiency (P450arom)

Aromatase (P450arom) is encoded by the CYP19 gene on chromosome 15p21.1, and aromatase deficiency was first described in Japan in 1991 [157] in a newborn with ambiguous genitalia. However, deficiency of this enzyme is very rare, and only a few rare mutations have been reported. Women who suffer from this deficiency experience, in the first instance, the impossibility of achieving normal levels of aromatization in the placenta, which allows the accumulation of the fetal androgen precursors that are used for the placental synthesis of estrogens. Therefore, this entity is associated with virilization of the mother in the second half of pregnancy, low maternal levels of estrogens, and a neonate that, if female, is masculinized [158]. Accurate prenatal diagnosis requires a DHEA and DHEA-S stress test. In a patient with placental sulfatase deficiency, estrogen levels increase in response to DHEA and not DHEA-S; meanwhile, patients with aromatase deficiency do not respond to any steroids. In other women, pubertal development does not occur because the ovary cannot aromatize androgens to produce estrogens. Therefore, the patient may present with primary amenorrhea (i.e., hypergonadotropic hypogonadism) and mild virilization. Recently, Praveen et al. [159] pointed out that although aromatase deficiency generally manifests at birth in females, the diagnosis may also be made later, either by abnormal pubertal development or by ovarian torsion due to (poly)cystic ovaries.

### 5.3. Congenital Hypogonadotropic Hypogonadism (Kallmann Syndrome) 

Kallmann syndrome is a genetic disorder or congenital form of hypogonadotropic hypogonadism (CHH) that manifests in hypo- or anosmia [160]. Although it is always mentioned in the literature as described by Kallmann et al. in 1944 [161], it was first referred to by the anatomical Spanish A. Maestre de San Juan in 1856 when describing “Teratology: Total lack of olfactory nerves with anosmia in an individual in who has congenital atrophy of the testicles and virile member” [162]. We currently know of six genes that are considered causal: FGF8, FGFR1, KAL1, PROK2, PROKR2, and CHD7; however, the sensitivity of molecular tests for CHH is only 30% [163]. Thus, diagnosis should be based on clinical findings including hypogonadotropic hypogonadism and anosmia or hyposmia, but depending on the genetic form of the disease also some non-reproductive and non-olfactory symptoms may be present. A very characteristic symptom is mirror movements of the upper extremities of the superior limbs (which mimic the synkinesis of the contralateral extremities) [163]. The condition is estimated to affect 1 in 48,000 individuals [164].

The symptoms are due to a failure in the differentiation or migration of neurons that arise embryologically in the olfactory mucosa, which are established in the hypothalamus and act as GnRH-secreting neurons [165]. The GnRH deficit results in decreased levels of sex steroids, which leads to a lack of sexual maturity and the absence of secondary sexual characteristics. So, the diagnosis typically occurs when a male does not enter puberty and shows lack of testicular development (as determined by testicular volume), or a failure of menarche (amenorrhea) in women. Poorly defined secondary sexual characteristics can include a lack of pubic hair and underdeveloped mammary glands. A small portion of males may present a micropenis, while cryptorchidism might have been present since birth. The condition is related to low levels of LH and FSH, which consequently results in low levels of T in men and estrogen and P in women [166]. There may be other associated malformations [167,168] and even unilateral renal agenesis [169]. Treatment involves lifelong hormone replacement therapy; and early surgical correction in undescended testicles [166]. These patients are at increased risk of developing osteoporosis later in life due to the decreased production of sex hormones, and will need to be prescribed vitamin D and bisphosphonate supplements [170].

## Figures and Tables

**Figure 1 jcm-09-03555-f001:**
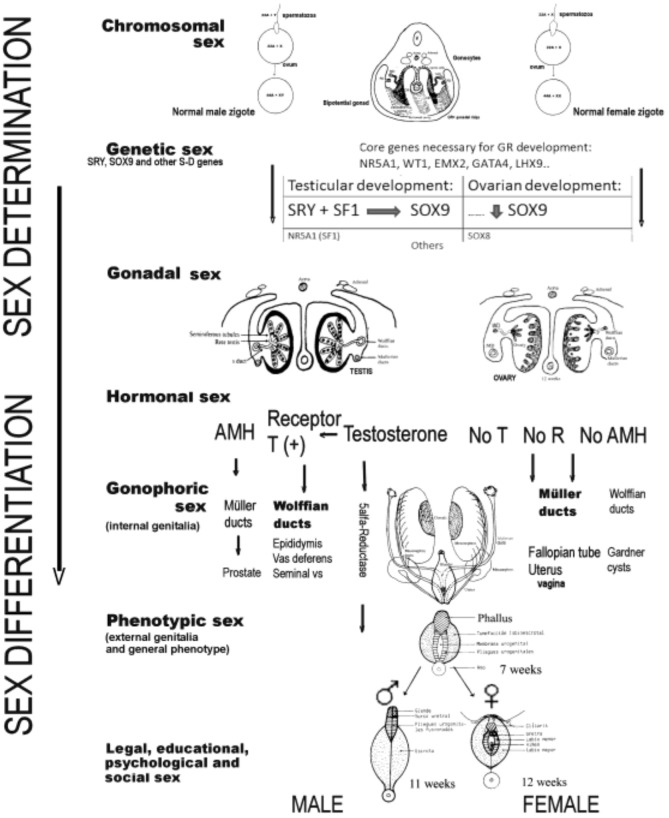
Processes and elements that intervene in the determination and differentiation of the human sex toward male or female (AMH, anti-Müllerian hormone; T, testosterone; R, receptors).

**Figure 2 jcm-09-03555-f002:**
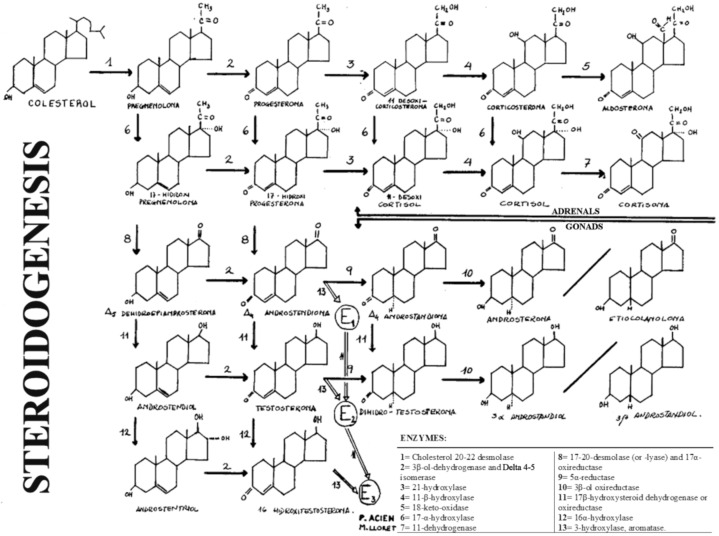
General diagram of steroidogenesis in the gonads and adrenal glands. “Reproduced with permission from Acién P, Tratado de Obstetricia y Ginecología: Obstetricia; published by Ed Molloy, Alicante, Spain, 1998”.

**Table 1 jcm-09-03555-t001:** Pathophysiological and clinical classification of the anomalies or disorders of sex development.

ANOMALIES IN SEX DETERMINATION (Chromosomes and Gonads) (Sex chromosome, 46,XY, or 46,XX DSDs in the Chicago Classification)		ANOMALIES IN SEX DIFFERENTIATION (Hormones and Enzymes) (Always 46,XX or 46,XY DSDs in the Chicago Classification)	
A. Without sex ambiguity (usually)	B. With sex ambiguity (usually)	A. Male Ps (non-dysgenetic Ps) or male (XY) with androgen deficiency (MAD) (if 46,XY and with testes)	B. Female Ps or female (XX) with androgen excess (FAE) (if 46XX and with ovaries)
1. Gonadal dysgenesis: (a) Ovarian agenesis (gonadal agenesis) (b) Gonadal dysgenesis with chromosomal and/or phenotypic alteration: Turner syndrome (c) Ovarian dysgenesis or hypoplasia	1. Testicular dysgenesis (XY): (a) Dysgenetic male Ps or partial gonadal dysgenesis. (b) Mixed gonadal dysgenesis	1. Gonadotropin-resistant testes and Fetal gonadotropic deficiency (Leydig cell hypoplasia)	1. Congenital adrenal hyperplasia in female/adrenogenital syndrome: (1) 17-α-hydroxilase deficiency (CYP17A1) (2) Cytochrome P450-oxidoreductase deficiency (PORD) (3) 21-hydroxylase deficiency (P450c21) (4) 11-β-hydroxylase deficiency (P450c11)
2. Variants of gonadal dysgenesis: (a) Mosaic variants(b) Structural abnormalities of the second sex chromosome(c) Pure gonadal dysgenesis. Swyer syndrome	2. True hermaphroditism or Ovotesticular disorders, OT-DSD	2. Deficiencies in the testicle itself or its secretions, including: (1) Embryonic testicular regression or testicular regression syndrome (anorquia) (2) Disorders of androgen production, i.e., male Ps or MAD due to blockage in steroidogenesis by enzyme deficiencies	2. Hormone therapy, iatrogenic
3. Triple X constitution and other polysomies	3. Mutations in the NR5A1 gene/SF-1	3. Defects in androgenic action or androgenic insensitivity syndromes (AIS) due to 5α-reductase deficiency or disorders in androgen receptor function	3. Maternal virilizing tumor
4. Sex reversal, males XX		4. Other mild forms of male Ps, or unambiguous (infertile male, cryptorchidism, hypospadias, persistent Müllerian duct syndrome).	4. Aromatase deficiency (P450arom)
5. Klinefelter syndrome, males			
6. Dysgenetic infertility		C. Without Ps: Congenital hypogonadotropic hypogonadism (CHH) (Kallmann syndrome, males and females)	

Ps: pseudohermaphroditism; MAD: male with androgen deficiency; FAE: female with androgen excess; DSDs: disorders of sex development; OT, ovotesticular; AIS, androgenic insensitivity syndromes; CYP, Cytochrome P450; In bold font, the most frequent anomalies.

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
