# Peer review of "Disorders of Sex Development: Classification, Review, and Impact on Fertility"

_jcm, 2020, doi:10.3390/jcm9113555_

Round 1

Reviewer 1 Report

It is an interesting and extensive overview of disorders of sex differentiation/development based on the current literature. I have some minor comments: 

  • Fig. 1. Some letters are too small and illegible  
  • Fig. 2. Letters are handwritten and some illegible. English names of enzymes should be placed here i.e. reductase, not reductasa etc. Steroidogenesis, not esteroidogenesis 
  • New guidelines on Klinefelter syndrome by Zitzmann et al. (Andrology. 2020 Sep 22. doi: 10.1111/andr.12909) appeared. I suggest to add this publication as a reference in the chapter 4A.5  
  • Line 399. I suggest to remove “well-developed" in the sentence “Ovotestes are defined by the coexistence of well-developed testes and ovaries in the same individual..”, because ovotestes have never well-developed testicular part what is mentioned some lines below. 
  • Leydigism – whot does it mean? 
  • Noxa – what does it mean? 
  • Zigote – should be zygote 
  • Eunucoid – should be eunuchoid 

Author Response

Responses to R1: Thank you very much for your review of our work and your interesting comments.

  1. Fig 1. We have redone the Figure and eliminated much of the text.
  2. Fig 2. We have modified the Figure by rewriting Steroidogenesis and all the enzymes. This figure is taken from Acién P (27) and we keep the drawings and handwriting of the original.
  3. New guidelines on Klinefelter syndrome: This reference (now 67) and several comments on the guide have been included in section 4A.5
  4. "well-developed" is deleted in ovotestes
  5. Leydigism: this is the term used in the title of the original publication by Klinefelter et al and we keep it "".
  6. Noxa: causal agent
  7. Zygote: OK

8. Eunuchoid: OK

Reviewer 2 Report

The proposed review is interesting and is of clinical relevance. Anyway, there are few suggestions:

  1. the terms "pseudohermaphroditism" and "hermaphroditism" should be less frequently used, given the aim of the review described in the introduction;
  2. Page 8, lines 309-310:it should be highlighted that the early dignosis and cognitive support in patients affected by Klinfelter syndrome can improve their QI and school performances. The terms "mentally retarded" and "social maladjusted" are not appropriate in this context;
  3. Page 13-24: it is necessary to summarize the paragraphs about enzymatic disorders in order to ameliorate the overall quality of the paper;
  4. Figures and tables are not clear as they contain too much information. They should be re-edited and subdivided when the information presented are too many. Please make sure that all the words in the figures and tables are readable.

Author Response

Responses to R2: Thank you very much for your review of our work and your interesting comments.

  1. "pseudohermaphroditism" and "hermaphroditism": we have restricted the use of these terms and in any case replaced by the acronyms Ps and TH (true hermaphroditism)
  2. Klinfelter syndrome: we have deleted the paragraphs with terms "mentally retarded" and "social maladjusted", and we have revised the early diagnosis and cognitive support adding the current reference 67.
  3. Enzymatic disorders.- We have reviewed everything related to them and tried to eliminate some paragraphs. However, it is difficult to summarize them further without losing the important information that we want to convey in this review.

4. Figures and table: have been redone and clarified. Many details have been erased and we believe that they now provide better information. We have modified the Figure 2 by rewriting Steroidogenesis and all the enzymes, but this figure is taken from Acién P (27) and we keep the drawings and handwriting of the original.